# Gamma-Chain Receptor Cytokines & PD-1 Manipulation to Restore HCV-Specific CD8^+^ T Cell Response during Chronic Hepatitis C

**DOI:** 10.3390/cells10030538

**Published:** 2021-03-03

**Authors:** Julia Peña-Asensio, Henar Calvo, Miguel Torralba, Joaquín Miquel, Eduardo Sanz-de-Villalobos, Juan-Ramón Larrubia

**Affiliations:** 1Translational Hepatology Unit, Guadalajara University Hospital, E-19002 Guadalajara, Spain; julia.pena@edu.uah.es (J.P.-A.); hcalvo@sescam.jccm.es (H.C.); miguel.torralba@uah.es (M.T.); jmiquelp@sescam.jccm.es (J.M.); eduardos@sescam.jccm.es (E.S.-d.-V.); 2Department of Biology of Systems, University of Alcalá, E-28805 Alcalá de Henares, Spain; 3Section of Gastroenterology & Hepatology, Guadalajara University Hospital, E-19002 Guadalajara, Spain; 4Service of Internal Medicine, Guadalajara University Hospital, E-19002 Guadalajara, Spain; 5Department of Medicine & Medical Specialties, University of Alcalá, E-28805 Alcalá de Henares, Spain

**Keywords:** Hepatitis C virus, CD8^+^ T cell response, exhaustion, immune checkpoints, γ-chain cytokines, PD-1, PD-L1, IL-15, IL-7, IL-21, IL-2

## Abstract

Hepatitis C virus (HCV)-specific CD8^+^ T cell response is essential in natural HCV infection control, but it becomes exhausted during persistent infection. Nowadays, chronic HCV infection can be resolved by direct acting anti-viral treatment, but there are still some non-responders that could benefit from CD8^+^ T cell response restoration. To become fully reactive, T cell needs the complete release of T cell receptor (TCR) signalling but, during exhaustion this is blocked by the PD-1 effect on CD28 triggering. The T cell pool sensitive to PD-1 modulation is the progenitor subset but not the terminally differentiated effector population. Nevertheless, the blockade of PD-1/PD-L1 checkpoint cannot be always enough to restore this pool. This is due to the HCV ability to impair other co-stimulatory mechanisms and metabolic pathways and to induce a pro-apoptotic state besides the TCR signalling impairment. In this sense, gamma-chain receptor cytokines involved in memory generation and maintenance, such as low-level IL-2, IL-7, IL-15, and IL-21, might carry out a positive effect on metabolic reprogramming, apoptosis blockade and restoration of co-stimulatory signalling. This review sheds light on the role of combinatory immunotherapeutic strategies to restore a reactive anti-HCV T cell response based on the mixture of PD-1 blocking plus IL-2/IL-7/IL-15/IL-21 treatment.

## 1. Introduction

Hepatitis C virus (HCV) is a non-cytopathic, hepatotropic, ssRNA *Flaviviridae* virus able to induce a chronic hepatitis in 65–85% of infected individuals with around 80 million viraemic population worldwide and, it is also the leading cause of liver-related death [1]. Nowadays, the natural history of HCV infection has changed due to the discovery of an effective treatment based on direct-acting antivirals (DAA), capable of impairing HCV replicative machinery [2]. Anyhow, the lessons learnt from anti-HCV CD8^+^ T cell restoration could serve as proof of concept for other chronic non-cytopathic viral infections and cancer but, its use for anti-HCV therapy appears currently unlikely. Nevertheless, there are still a few cases with multiple viral resistances to DAA that could benefit from boosting the adaptive immune response [3].

HCV-specific cytotoxic T cell response is essential in natural HCV clearing [4,5], however this virus has developed several strategies to escape from CD8^+^ T cell control [6]. First of all, this virus has a great genetic variability due to the lack of proof-reading in its RNA-polymerase, leading to the rapid generation of escape-mutation variants [7]. Although there are conserved epitopes where are not possible these variants to occur due to the loss of viral fitness [8,9,10], the virus is also able to induce CD8^+^ T cell exhaustion [11,12] by affecting T cell receptor (TCR) triggering through induction of negative costimulatory molecules [13], in addition to changes in the survival mechanisms [14], mitochondrial reprograming [15,16], CD8^+^ T cell chemotaxis [17], impairment in CD8^+^ T cell co-stimulation [18], loss of CD4^+^ T cell help [19,20] or by induction of regulatory CD4^+^ T cells (Tregs) [21,22]. Therefore, strategies focused on dealing only with CD8^+^ T cell negative co-stimulation could not be enough to restore CD8^+^ T cell response, because although T cell signalling could be restored [13,23] there could be other T cell damages that should also be set up in order to improve T cell reactivity.

The CD8^+^ T cell differentiation program starts with a stem-cell like subset, called precursor pool that is able of self-renewal and it is in charge of giving rise to the effector progeny [24,25,26]. This specific pool is characterized by the expression of the specific transcription factor T cell factor 1 (TCF-1) and receptors for gamma-chain(γc) cytokines such as interleukin (IL)-7 receptor (R), IL-21R or IL-15R [27,28]. This precursor pool sustains CD8^+^ T cell immune response during persistent viral infection and displays markers of T cell exhaustion, such as programmed cell death protein-1 (PD-1) up-regulation but it is also the subset sensitive to PD-1 modulation [29].

This review sheds light on the mechanisms involved in HCV-specific CD8^+^ T cell exhaustion and how strategies directed to improve TCR signalling in the precursor pool might be enough in the initial stage of the infection. In more progressed disease will be necessary to carry out combination strategies, in which the addition of survivals cytokines could be effective [30,31], and finally in long-term disease all these approaches would fail due to the probably deletion of these specific CD8^+^ T cells [32].

## 2. Key Role of Specific CD8 T Cells in HCV Infection

HCV develops several mechanisms to escape from host’s innate and adaptive defences, such as interferences of HCV with endogenous type I interferon (IFN) produced by infected cells, impairment of plasmacytoid and conventional dendritic cells, induction of natural killer cell defects, lack of neutralizing antibodies, induction of regulatory cytokines (IL-10, tumour growth factor (TGF)-β1), induction of Tregs, lack of CD4^+^ T cell help and CD8^+^ T cell response evasion [33,34]. To escape from HCV-specific CD8 T cell response is a key action in HCV survival. The cytotoxic T lymphocytes can remove HCV by cytolytic and non-cytolytic mechanisms [35]. In infected humans and chimpanzees, there is a clear temporal association between HCV clearance and the appearance of IFN-gamma(γ) secreting HCV-specific CD8^+^ T cells [4,5,36,37]. In vivo, CD8^+^ T cell depletion studies show that is not possible to achieve HCV clearance without these cells [38]. HCV, as a non-cytopathic virus, needs the long-term host survival to increase its chance of being transmitted to another subject. Therefore, a fine-tuned evolutionary balance has arisen between HCV and specific CD8^+^ T cell response. HCV has developed different strategies to escape from CD8^+^ T cell control. The lack of proof-reading mechanism leads HCV to develop escape mutations under immune selection pressure in TCR recognition sites [7,39,40]. In fact, in those subjects with haplotypes presenting HCV epitopes with low mutation potential, it is more prevalent the presence of spontaneous virus clearance [8,9,10,41]. Nevertheless, the occurrence of this HCV variants is not always possible, since stable cytotoxic T cell escape mutation in hepatitis C virus is linked to maintenance of viral fitness [39,42]. Besides this persistence strategy, HCV is also able to induce exhaustion and apoptosis on those CD8^+^ T cells still able to recognize HCV epitopes [14,43]. The exhausted HCV-specific cytotoxic T cells during chronic HCV infection display low proliferation span after antigen encounter and low ability to secrete type I cytokines and they are prone to apoptosis [14,44,45,46]. These exhausted cells are able to maintain a persistent low-grade control of HCV but without the possibility of viral clearance. This incomplete viral control sustains a persistent release of proinflammatory cytokines and chemokines from the infected liver that attracts non-specific inflammatory cells responsible of the persistent low-level liver damage [17,47]. Combining both escape mutations and CD8^+^ T cell exhaustion, HCV is capable to persist in around 60–80% of infected patients [48]. From the bulk repertoire of HCV-specific CD8 T cells during persistent infection, around 50% develops escape mutations while the another 50% displays exhaustion features, characterized by the expression of different negative immunoregulatory checkpoints (IC) [18].

T cell exhaustion is a progressive loss of effector function due to prolonged antigen stimulation, characteristic of chronic infections and cancer. During the exhaustion process, a sequential loss of effector functions occurs. After a long-lasting viral infection, IL-2 secretion and the ability to lyse target cells are the first CD8^+^ T cell functions deleted, followed by the suppression of tumour necrosis factor (TNF)α and IFNγ secretion and a subsequent final deletion of T cells (Figure 1). Antigen appeared to be the driving force for this loss of function, since a strong correlation exists between the viral load and the level of exhaustion [12]. Globally, the speed of this process depends on the level of antigen exposure and the duration of infection [32].

The exhausted effector pool in charge of controlling HCV replication during persistent infection is characterized by, besides the expression of several negative IC [18,49], the lack of receptors for survival γc cytokines [43,46]. In this subset, the expression of some of these negative checkpoints, such as PD-1, are very high, and they are not sensitive to immunotherapeutic strategies [25]. This effector pool shows an anabolic metabolism with low level of mitochondrial fatty oxidation and high production of reactive oxygen species that makes these cells prone to apoptosis [50]. These cells keep initially their killing abilities before approaching a late dysfunctional state, featured by upregulated expression of negative co-stimulatory molecules, such as T cell immunoglobulin domain and mucin domain (Tim)-3 or cytotoxic T-lymphocyte–associated antigen (CTLA)-4, and CD39, which defines an irreversible level of exhaustion [27]. The hallmark of this population is the expression of the transcription factor High-Mobility Group (HMG)-box protein TOX, which is essential for the establishment of exhaustion [51]. This effector population is generated continuously from the precursor pool. During differentiation of the precursor subset have been described four steps [27] (Figure 2). The first stage comprises TCF1^high^ stem-like precursor cells that are quiescent and tissue resident; the next differentiation step includes TCF1^+^ precursor cells that are detected in the periphery and with capacity to proliferate; the third differentiation level gives rise to an effector-like TCF1^−^ transitory population, which relies on IL-21 for its formation and expresses the chemokine receptor CX3CR1 [52]; finally a late dysfunctional TCF1^−^ population is generated, which expresses high levels of negative IC and lacks proliferative capacity [27].

The precursor pool has self-renewal potential but it is also able to give rise to the effector pool [24]. They have a catabolic metabolism with mitochondrial fatty acid oxidation and oxidative phosphorylation [50]. This subset expresses the negative co-stimulatory checkpoint PD-1 but to a lower level than the effector pool and, do not express other negative checkpoints such as Tim-3 [27]. It is noteworthy to know that the precursor PD-1^dim^ pool is characterized by being resistant to the DNA damage produced during the exhaustion process in the more differentiated T cells [53], converting them in an interesting target for immunomodulatory strategies. This TCF-1-expressing precursor CD8^+^ T cell pool develops a key role in viral control, since it sustains the immune response to chronic viral infections, such as HCV. During a chronic viral infection, the presence of TCF-1^+^ viral-specific CD8 T cells promotes the effector functions of exhausted CD8^+^ T cells [28]. This precursor population shows memory markers such as common γc cytokine receptors that includes IL-7R, IL-21R and IL-15R [27,54,55]. The intermediate PD-1 level of these precursor cells makes them sensitive to PD-1/PD-ligand(L)1 blockade [25,29], being the likely target of current anti-PD-1 oncologic treatments, such in hepatocellular carcinoma [56]. Negative IC blockade could be combined with the addition of survival γc cytokines to potentiate the effects of PD-1 blockade on T cell exhaustion. The exhaustion profile is fixed epigenetically and limits the duration of reinvigoration by PD-1 blockade [57,58], but we could perhaps surpass this effect by adding the immune-modulatory properties of the γc cytokines [30,59].

## 3. PD-1 Modulation for T Cell Exhaustion Reversion

PD-1 is the negative IC more widely studied to revert the effector cytotoxic T cell response against viral infection and cancers. Although PD-1 is a marker of exhaustion on T cells, it is also a marker of T cell activation. Dysfunctional CD8^+^ T cells and activated CD8^+^ T cells up-regulate genes involved in activation of the cell cycle, T cell homing and migration, as well as effector molecules, such as granzymes and co-stimulatory and co-inhibitory receptors [60]. In fact, the PD-1^high^ expression on HCV-specific T cells during acute infection do not correlate with the clinical outcome, suggesting that the PD-1 level is marking activation but not exhaustion during acute hepatitis [61]. Currently, there are available several molecules to block both PD-1 and its ligands that are being utilized to treat different types of advanced cancers [62]. PD-1 suppresses T cell function by preferentially dephosphorylating CD28 [23], which in turn is needed for the recovery of T cells subjected to anti-PD-1 immunotherapy [13], (Figure 3). CD28 is a positive co-stimulatory signal involved in naïve and memory T cell activation after TCR triggering. The PD-1^dim^ stem-like precursor T cells express CD28, making these cells sensitive to PD-1 modulation. Thereafter, the generated progeny loss the expression of this receptor, making impossible the response to PD-1 blocking [63]. Besides this CD28 dependent PD-1 inhibitory mechanism, it has been recently reported that PD-1 can also inhibit T cell activation upon TCR triggering in absence of CD28 co-stimulation [64]. Therefore, PD-1 blocking could be useful as a short-term strategy to rescue CD28 negative effector cells but, to obtain a more persistent response it will probably necessary to restore the functionality of CD28 positive memory T cells. Another issue, regarding the role of CD28 in PD-1 blockade, is related with the regulatory properties of T regs. These cells are enriched in HCV infection [65] and they express CTLA-4 that can degrade CD80 and CD86 ligands [66], which could impair CD28 co-stimulation and potentially decrease the efficacy of PD-1 blockade. In this line, HCV-specific CD8^+^ T cells derived from liver biopsies of chronically HCV-infected patients required CTLA-4 blockade in addition to PD-1 blockade to restore their function [67]. In those exhausted CD28 negative effector CD8 T cells, other co-stimulatory molecules from the tumour necrosis factor receptor (TNFR) family such as tumour necrosis factor receptor superfamily member 9 (4-1BB), glucocorticoid-induced tumour necrosis factor receptor family–related protein (GITR), tumour necrosis factor receptor superfamily member 4 (OX40) or CD27 could be expressed to sustain T cell activation, although despite the presence of these positive IC, these cells become finally dysfunctional during persistent infection [32,68,69].

Several studies have linked the PD-1 up-regulation with the development of HCV persistence [70,71] and have suggested a potential role of PD-1 blockade as strategy for HCV control [43,72]. The effect of PD-1 blocking on HCV-specific CD8 T cells has been widely studied both in-vitro and in-vivo. The response to PD-1 blockade depends on the level of PD-1 expression and the compartment of treated T cells. The intrahepatic HCV-specific CD8 T cells display high PD-1 level [46] and lack of CD28 expression, which as a result could make those cells less sensitive to anti-PD-1 treatment [45]. Despite this drawback, the blockade of PD-1/PD-L1 pathway has been used to treat chronic HCV patients as proof of concept with promising results [73]. In this randomized, double-blind, placebo-controlled assessment of a fully human monoclonal antibody against PD-1, a significant decrease in HCV viral load was observed and one case remained HCV negative during follow-up after treatment. Nonetheless, the in-vitro studies showed that bulk of cases failed to respond to the single blockade of the PD-1/PD-L1 pathway. Therefore, additional strategies, especially combination therapies, has been initiated by adding the blockade of other regulatory checkpoints. Several blocking combinations of negative co-stimulatory checkpoints have been tried. Overall, strategies combining PD-1 plus either 2B4, Tim-3 or CTLA-4 blockade in peripheral cells yielded three response possibilities: total non-response, good single blockade response and good dual/multiple blockade response, with each comprising approximately one-third of the patients tested [74]. The combined blockade of CTLA-4 and PD-1 has shown to be also efficient in the restoration of intrahepatic CD28-expresing HCV-specific CD8 T cells, which are more exhausted than the peripheral pool [67]. Another synergic checkpoint modulation is based on the stimulation of the late positive co-stimulatory receptor 4-1BB in addition to PD-1 blockade. Combined blockade of PD-1 plus 4-1BB stimulation increased responses of intrahepatic T cells against HBV, but not HCV [75]. This failure in hepatis C was due to the loss of the 4-1BB signal transducer, but after inducing the expression of TRAF-1 in these cells, the PD-1 blockade plus 4-1BB stimulation was also able to restore HCV-specific CD8 T cell reactivity, as it will be discussed later [32].

This in-vitro strategies are efficient in increasing functionality of HCV-specific cytotoxic T cells, but probably are not long-lasting due to the epigenetic stability of the exhaustion pathways on exhausted T cells, which will transfer the exhausted phenotype to their progeny [57]. Under these circumstances, to maintain a chronic treatment with negative IC blocking antibodies would be necessary with the associated risk of becoming resistant to the treatment. Another approach could be to combine the PD-1 blockade to restore TCR signalling with strategies directed to rejuvenate these cells in order to delete the epigenetic imprints due to the previous exhaustion process. In this sense, γc cytokines could have a positive role since PD-1 intermediate precursor subset display the appropriate receptor to be sensitive to the positive effects of these cytokines on survival and metabolism. 

## 4. Gamma (γ) Chain Cytokines for T Cell Exhaustion Reversion

The common γ-chain family of cytokines includes IL-2, IL-7, IL-15 and IL-21. Each cytokine binds its specific heteromeric receptor composed of two or three chains: a specific α chain, the common γ chain and the β chain for IL-2 and IL-15 [59]. Although IL-2 supports mainly effector T cell differentiation, IL-7, IL-15 and IL-21 are involved in the development and expansion of the PD-1^dim^ CD28^+^ precursor T cell subset [27]. In the next sections, this review will focus on the role of possible synergic combinations of IL-2, IL-7, IL-15 and IL-21 plus PD-1 blockade to surpass the exhaustion status of the stem-like precursor subset. In fact, in the natural development of stem-like precursor cells to give rise to the progeny, IL-21 secreted by CD4 T cells is needed [52]. Moreover, the expansion of Ag-specific T cell in presence of IL-21, IL-15 or IL-7 is superior to IL-2 in driving less differentiated T cells with precursor phenotype and longer survival [76]. The importance of γc cytokines in stem-like precursor pool is highlighted by intense expression level of the β-chain of the IL-2 and IL-15 receptor (IL-2Rβ) in these cells [54].

### 4.1. Interleukin-2

IL-2 is a key cytokine that regulates clonal expansion and effector/memory differentiation of CD8 T cell after antigen encounter. This cytokine is mainly produced by CD4 T cells, but also by CD8 T cells after activation. The IL-2R is comprised by three subunits: IL-2Rα (CD25), IL-2Rβ (CD122) and IL-2Rγ (CD132, γc). CD25 is not constitutively expressed on resting CD8 T cells, but it is up-regulated after TCR triggering [77]. IL-2 is the third essential signal, in addition to TCR triggering plus positive co-stimulation, to promote cellular proliferation during acute infection and, it is also essential for the development of potent secondary CD8^+^ T cell responses [78]. This cytokine has a key role as a differentiation factor leading to the generation of terminally differentiated effector CD8 T cells and decreasing the number of memory precursors [79]. This effect is promoted by the IL-2 induced expression of the transcription factor Blimp-1 [80]. It seem that autocrine IL-2 secretion by CD8 T cells could favour the generation of memory CD8 T cells but paracrine IL-2 derived from CD4 T cells gives rise to terminally differentiated effector cells [81]. The IL-2 feature to promote memory cells when is present in low level could be useful to rescue viral-specific CD8 T cells during persistent viral infections. However, its capacity to promote T cell terminal differentiation and exhaustion is a critical limiting factor that must be carefully considered. Moreover, IL-2 induces Tregs generation, which could promote the exhaustion persistence on CD8^+^ T cells [82].

### 4.2. Interleukin-7

IL-7 receptor is a heterodimer formed by the γc plus the IL-7Rα chain (CD127). This receptor is a marker for the subset of T cells committed to become memory cells [83]. IL-7 supports primmed T cell survival through the expression of anti-apoptotic molecules, such as Mcl-1 or Bcl-2 in a STAT5 dependent manner [84]. Moreover, IL-7 allows the homeostatic maintenance of memory T cells in those situations lacking of the persistent cognate antigen [85]. IL-7 can also limit the development of T cell exhaustion during chronic antigen stimulation [86]. In this sense, IL-7 decreases the PD-1 expression in chronic viral infection models [87]. IL-7 downregulates a repressor of cytokine signalling, resulting in amplified cytokine production and increased T cell effector functions [30]. The combination of IL-7 plus IL-15 treatment promotes the development of long-lasting stem-like cells [55], which could be interesting as strategy to restore an effective T cell response. In chronic hepatitis C the expression level of IL-7R correlated with the balance between the pro- and anti-apoptotic molecules Bim and Mcl-1 on HCV-specific CD8 T cells. CD127^low^ cells up-regulated Bim after Ag encounter, which leaded them to apoptosis [14]. Besides this antiapoptotic effect, IL-7 has also shown an effect to revert HCV-specific CD8 T cell exhaustion. In-vitro IL-7 treatment was able to restore T cell reactivity by making these cells sensitive to 4-1BB co-stimulation. In-vitro IL-7 treatment is able to upregulate TRAF1 expression, making 4-1BB signalling active and restoring T cell reactivity [32]. All these features make this cytokine a clear candidate to manipulate the exhausted PD-1^dim^ precursor pool, which is the current target to achieve a persistent viral control.

### 4.3. Interleukin-15

IL-15R is comprised of IL2Rβ, γc and a specific α-chain (IL15Rα, CD215). IL-15 is usually bound to the alpha chain (IL-15Rα) and it is trans-presented to the IL2Rβγ heterodimer on T cells to mediate its biological activity [88]. IL-15 preserves the homeostatic maintenance of memory T cells and promotes T cell survival by induction of anti-apoptotic molecules. IL-15 not only plays an important role in the maintenance of the CD8 memory subset [89], but this cytokine is also key in dictating the composition of the specific T cell pool. Lacking IL-15, memory CD8 T cells have a reduced cell cycle and impaired Bcl-2 expression, suggesting a role for IL-15 in supporting basal proliferation and survival of memory cells. Besides, IL-15 deficiency results in absence of CD127^low^ memory cells and in changes within the CD127^high^ CD62L^low^ memory pool, which expresses high levels of CD27 and low granzyme B [90]. Therefore, IL-15 not only maintains memory response but also regulate its composition. Moreover, the rapid loss of IL-2 secretion by T cells during chronic viral infections [91] could convert IL-15 in a key actor during effector CD8 T cell differentiation. These features make IL-15 better than IL-2 to develop Ag-specific T cells, because both can induce T cell expansion but IL15 does not promote the terminal differentiation, preserving the memory phenotype and improving the metabolic profile of the generated cells [84,92]. Moreover, IL-15 favours T cell self-renewal [54], which is another key characteristic of the precursor stem-like pool. On top of all these positive actions, IL-15 also induces a catabolic metabolism linked to autophagy, self-renewal and asymmetric cell division [24,50]. Mitochondrial dynamics controls T cell fate through metabolic reprogramming and, consequently the induction of fused mitochondria, with tight cristae and efficient oxidative phosphorylation would lead to non-exhausted memory cells [93]. T cells developed in a rich IL-15 environment promotes mitochondrial biogenesis with increased fatty-acid oxidation and high spare respiratory capacity [94]. On the contrary, PD-1 is an early driver of T cell exhaustion by the induction of bioenergetic insufficiencies due to metabolic alterations [95] that could be counteract by IL-15 treatment. Another IL-15 positive effect is the survival support on T cells by the regulation of anti- and pro-apoptotic molecules, such as Bim and Mcl-1 [96]. Bearing in mind that the target population to revert T cell exhaustion in chronic HCV infection is the PD-1^dim^ memory-like precursor pool and that IL-15 promotes the development of these cells and improves their metabolic profile, the combination of PD-1 blockade plus IL-15 could be an interesting strategy to get this goal. This combination has shown to enhance CD8 T cell function in chronic viral infections [97,98] and also has recovered tumour-infiltrating PD-1 unresponsive CD8 T cell due to loss of CD28 [99]. Moreover, the development of CAR-T cells within a rich IL-15 environment augments the response to PD-1 blockade in solid tumours [100].

### 4.4. Interleukin-21

IL-21R comprises the IL21Rα chain and de γc and it is expressed by CD8 T cells and up-regulated after TCR triggering [101]. IL-21 induces the expression of several transcription factors, such as T-bet, Eomes, Bcl-6, Blimp-1, BATF and IRF4 [102]. This cytokine is involved in CD8 T cell differentiation but it depends on the T cell context [101]. IL-21 can induce antigen-specific proliferation and effector abilities but also can promote less differentiated memory-like T cells by inducing the expression of transcription factors such as TCF-1 and Bcl-6 [103]. Moreover, in a similar manner to IL-15, IL-21 induces a mitochondrial reprogramming on CD8 T cells from an anabolic profile to a more catabolic one, featured by fatty acid oxidation and oxidative phosphorylation [104], which promotes the development of memory-like precursor cells [105]. IL-21 is also involved in decreasing the exhaustion level by reducing PD-1 expression [104]. In fact, during the differentiation process of virus-specific CD8 T cells, IL-21 carries out a key role in the development of the stem-like precursors that will give rise to the precursor pool and in the generation of the effector-like transitory subset. Therefore, IL-21 is involved in generation of both the precursor and the progeny pools [27]. These interesting properties converts this cytokine as a potential tool to generate non-exhausted precursor cells in chronic viral infections. In persistent viral infection models, IL-21 rescues the effector ability of virus-specific CD8 T cells by sustaining the expression of the transcription factor Blimp-1 [106]. Besides this effect on effector cells, IL-21 in tumour models has also increased the long-term survival of functional memory cells [107]. This effect on memory cells is synergized by its combination with IL-15 [108] and the blocking of negative IC, such as PD-1 or CTLA-4 [109]. Therefore, IL-21 shows a clear benefit in boosting effector T cells but also in leading to generate precursor stem-like cells. Table 1 summarises the main actions of γc cytokines on memory-like CD8^+^ T cells.

## 5. Effect of PD-1 Modulation and γc-Cytokines on HCV-Specific CD8^+^ T Cell Response

HCV-specific CD8 T cells targeting HCV epitopes during persistent infection become exhausted and expresses several negative IC, such as PD-1 [18]. Different in-vitro studies have shown that the PD-1/PD-L1 blockade can restore T cell reactivity in some patients [43,72] and, there are also clinical evidences that blocking this pathway could lead to HCV control [60,73]. Nevertheless, a bulk of HCV patients are not responders to this immune modulation. Currently, we know that the target population sensitive to PD-1 blockade is the PD-1^dim^ precursor memory-like subset in charge of giving rise to the exhausted progeny [25]. In fact, the PD-1^high^ HCV-specific CD8 T cells are refractory to PD-1/PD-L1 blockade [45]. The precursor memory-like pool has been widely described in chronic hepatitis C and it is characterized by PD-1^dim^ and CD127 expression [26,29,110]. Unfortunately, releasing the CD28 co-stimulation after PD-1 blockade will rescue the normal TCR triggering [13,23], but this will not be enough in most of the exhausted T cells. After a short-term infection, exhausted HCV-specific T cells will have developed an epigenetic imprint that will induce the sustained impairment of several cellular functions [57], such as the development of an anabolic-prone metabolism [15,50], the impairment of positive co-stimulation [32] or induction of a pro-apoptotic state [14]. Therefore, to restore an efficient T cell response, the blockade of negative IC plus the boost of a catabolic mitochondrial metabolism, the improvement of positive co-stimulation and counteracting pro-apoptotic status should be combined. Strategies focused on mitochondrial reprogramming to increase fatty acid oxidation and oxidative phosphorylation could lead to give rise to a restored precursor stem-cell-like subset [93,94,105]. For these immunotherapeutic approaches γc-cytokines could be an excellent candidate due to their pleiotropic effects on metabolism, survival and co-stimulation, as previously discussed [100,104].

The role of IL-2 treatment in combination with PD-1 blockade has not been tested in chronic HCV infection. Nevertheless, in the mouse model of chronic lymphocytic choriomeningitis virus (LCMV) infection, low-dose of IL-2 combined with PD-1 blockade enhanced CD8^+^T cell response, leading to viral load decrease. IL-2 decreased the level of inhibitory receptors and increased the expression of CD127, suggesting a role in memory generation [82]. Therefore, low IL-2 dose plus anti-PD-1 treatment should be considered to rescue exhausted T cells in chronic viral infections and cancers. Nevertheless, CD8 T cell treatment with high dose of IL-2 promotes the differentiation into effector/exhausted phenotype, while treatment with IL-7 or IL-15 gives rise to TCF1^+^ cells that are highly responsive to anti-PD-1 blockade [100]. During hepatitis C virus infection IL-7 and IL-15 have shown to increase the antiviral efficacy of CD127-positive but not of CD127-negative, HCV-specific CD8 T cells [111]. These data suggest that these γc-cytokines could influence the restoration of the precursor pool, which are featured by the expression of the IL-7R [25]. In fact, tumour infiltrating CD8^+^ T cells are TCF1 negative and CD28 negative and do not respond to PD-1 blockade. Nevertheless, these cells still responds to IL-15 treatment, which makes them sensitive to anti-PD-1 treatment and enhanced their proliferation, probably by restoring the precursor pool [99]. Nevertheless, HCV downregulates interferon regulatory factor-2 in hepatocytes, attenuating hepatocellular expression of IL-7 and IL-15Rα to abrogate the possible benefit of these homeostatic cytokines on HCV-specific T-cell responses [112]. Also, a better anti-viral CD8 T cell response and HCV control are associated with increased IL-21 levels. A low frequency of IL-21-secreting CD4 T cells, correlates with an up-regulation of inhibitory receptors, such as CTLA-4, PD-1 and Tim3, while IL-21 in-vitro treatment of HCV-specific CD8 T cells enhanced their proliferation and prevented Galectin-9 induced apoptosis [113]. According to these observations, a potential strategy to rescue HCV-specific CD8 T cell response would be the synergy between homeostatic γc-cytokines, such as low level IL-2, IL-7, IL-15 or IL-21, plus PD-1 blockade.

This kind of combinations are being currently explored in preclinical cancer models with either IL-15 [114,115] or IL-7 [116] or IL-21 [109]. Nevertheless, to our knowledge, there is only one in-vitro study testing the effect of PD-1 plus γc-cytokines on HCV-specific CD8 T cells [32]. This work shows the synergic effect of anti-PD-L1 plus IL-7 to restore HCV-specific CD8 T cell reactivity after Ag-specific in-vitro challenge. During HCV infection there is a hierarchical exhaustion gradient according to the extension of infection and the rate of liver fibrosis progression. In short-lasting HCV infection and low-rate liver fibrosis progression cases, the treatment with IL-7 was enough to restore T cell reactivity. In patients with an intermediate stage with either long-lasting disease or rapid hepatic fibrosis rate, the anti-PD-L1 plus IL-7 treatment rescue T cell reactivity. Finally, in more advanced cases with both long-lasting disease and rapid rate of liver fibrosis, T cell reactivity was not restored with any treatment combination, probably due to the loss of these T cells by apoptosis (Figure 4), as previously described in the LCMV infection [91]. In this work, it is also described an IL-7 mechanism to explain the machinery involved in T cell reactivity restoration. Previously, to restore intrahepatic PD-1^high^ HCV-specific CD8 T cell reactivity by 4-1BB stimulation plus anti-PD-L1 treatment had been unsuccessful. This failure was probably due to the loss of 4-1BB signal transducer TRAF-1, promoted by the HCV-induced TGF-β1. Interestingly, IL-7 counteracted the TGF-β1 effect, making these cells sensitive to the 4-1BB positive co-stimulation. After restoring 4-1BB signalling, these cells could also benefit from the blockade of the PD-1/PD-L1 pathway (Figure 4). Similar observation has been described in human immunodeficiency virus infection (HIV). In this human viral chronic infection, HIV-specific CD8 T cells also lose the 4-1BB transducing factor (TRAF1), which is re-expressed by the action of IL-7, linked to enhancement of T cell reactivity [68]. This two works suggest a general viral mechanism to promote exhaustion by impairing the 4-1BB positive co-stimulation through TGF-β1 effect that can be counteract by IL-7 treatment. Anyhow, other potential effects of IL-7 on these cells cannot be excluded, such as the anti-apoptotic role of this cytokine [14]. This work is a proof of concept of the high immunotherapeutic potential of combining γc-cytokines plus PD-1/PD-L1 pathway blockade to rescue HCV-specific CD8 T cells from exhaustion, but it is also necessary to address the role of IL-15 or IL-21 plus anti-PD-1 treatment in HCV infection.

Nonetheless, most of the data about the effect of these cytokines on CD8 T cell response restoration have been generated in-vitro and in animal models. Before the clinical application of this combination therapies the potential toxicity should be addressed. Most of the information about γc cytokine and anti-PD-1 treatment toxicity has been obtained from cancer clinical trials. The single PD-1/PD-L1 blockade can induce autoimmune-like/inflammatory side-effects that can be usually clinically controlled [117]. IL-15 at high dose regimen is associated with hypotension, thrombocytopenia, hepatitis and increase of several pro-inflammatory cytokines [118]. IL-2 treatment can induce grade 3–4 toxicity, including hypotension, cardiac ischemia, vascular leak syndrome and sepsis [119]. IL-7 has been tested in phase 1 clinical trials in humans with acceptable tolerance [120]. In any case, the use of this combinatory treatments out of conditioning strategies for adoptive cell transfer must consider the balance between the benefit and the potential autoimmune or pro-inflammatory adverse events in the host.

## 6. Conclusions

The restoration of exhausted HCV-specific CD8 T cells must be based on targeting PD-1^dim^ precursor pool. This subset is sensitive to PD-1/PD-L1 blockade, but this strategy cannot be enough in most of the cases due to the steady epigenetic changes induced by the exhaustion in different cellular functions, such as mitochondrial metabolism, positive co-stimulation or pro-apoptotic status. The addition of the homeostatic pleotropic γc-cytokines to the PD-1 pathway blockade could counteract the metabolic changes and the co-stimulation failures. At least the synergic effect of anti-PD-L1 plus IL-7 treatment has been successful in restoring HCV-specific CD8 T cell reactivity in vitro. It is mandatory for future investigations to analyse the role of IL-15/IL-21 plus PD-1 blockade, due to the specific mitochondrial reprograming induced by these cytokines. Although, these strategies are currently unlikely to be applied for HCV control due to effective DAA therapy, they can be used as proof of concept for other chronic viral infections and tumours.

## Figures and Tables

**Figure 1 cells-10-00538-f001:**
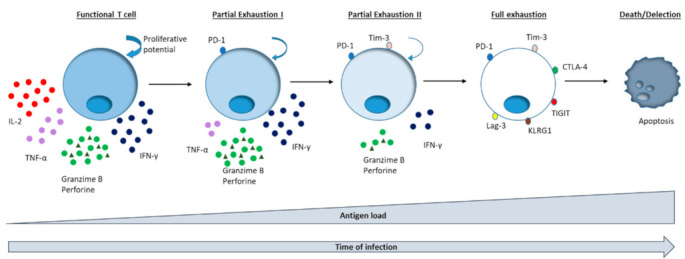
Scheme representing the hierarchical loss of effector functions on exhausted antigen specific CD8^+^ T cells during persistent viral infection.

**Figure 2 cells-10-00538-f002:**
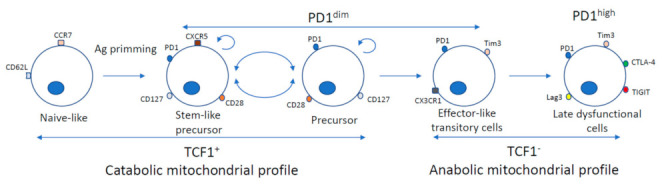
Differentiation program of CD8 T cells during chronic HCV infection. Naïve-like T cells expresses the transcription factor TCF1 and the lymphocyte homing and adhesion molecule receptors CD62L and CCR7 and, the positive co-stimulatory molecule CD28. After T cell priming, the naïve pool differentiates into stem-like precursor and precursor subset. These pools maintain TCF1 expression, are CD127 positive, display a catabolic mitochondrial metabolism and have self-renewal potential. The stem-like precursor subset expresses the chemokine receptor CXCR5 and are found in primary lymphoid organs. The precursor pool loses CXCR5 and can be found in the periphery, giving rise to the effector-like transitory subset, but also can dedifferentiate into the stem-like precursor pool. The progeny subset is comprised by the effector-like transitory cells, which expresses the chemokine receptor CX3CR1 and loss TCF1 expression. Finally, this pool differentiates into the late dysfunctional subset characterized by loss of functionality and the expression of several negative immune checkpoints.

**Figure 3 cells-10-00538-f003:**
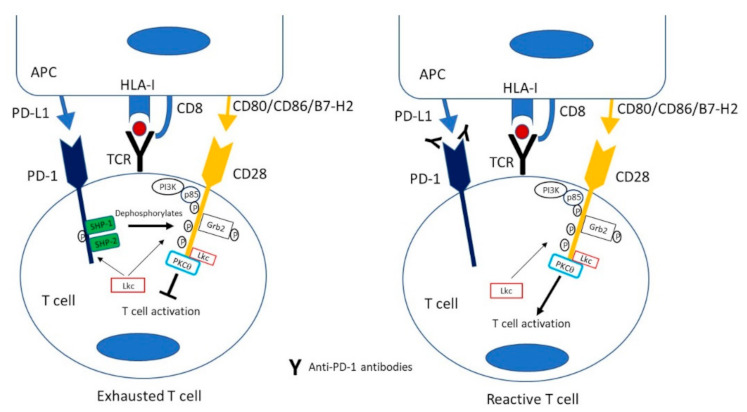
PD-1 & CD28 signalling in exhausted T cells and effect of PD-1/PD-L1 blockade. Src kinase Lck phosphorylates tyrosine residues on PD-1 and CD28. This allows CD28 to recruit PI-3K and GRB2 and allows PD-1 to recruit phosphatases SHP-2 and SHP-1. SHP-2 dephosphorylates CD28 attenuating the signalling via CD28. Antibodies against PD-1 or PD-L1 hamper PD-1 activation, releasing CD28 co-stimulation. APC: antigen presenting cell. TCR: T cell receptor. P: phosphorylated, PD-1: Programmed cell death protein 1, PD-L1: Programmed Death-ligand 1.

**Figure 4 cells-10-00538-f004:**
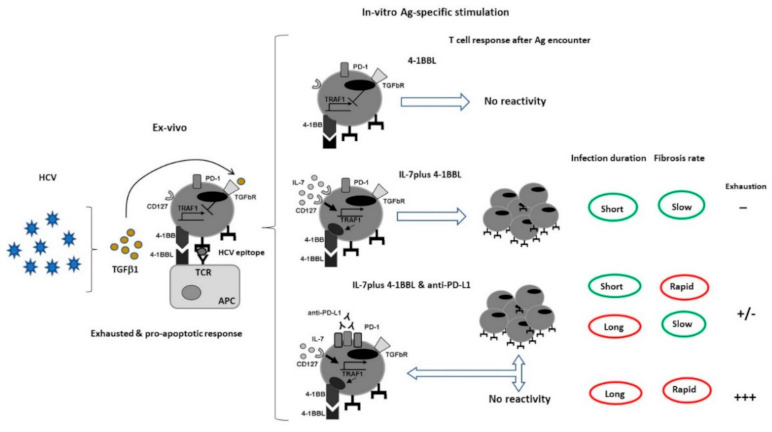
IL-7 effect on restoring 4-1BB signal transducer in HCV-specific CD8 T cells. TGFβ-1 induced by HCV infection promotes TRAF1 loss avoiding 4-1BB signalling. IL-7 counteracts this effect making these cells sensitive to 4-1BB triggering. This treatment restores reactivity in the less advanced cases. Those more progressed patients also need the PD-1/PD-L1 pathway blockade. Finally, patients with long-lasting disease and rapid rate of liver fibrosis are not restorable. 4-1BB: tumour necrosis factor receptor superfamily member 9, 4-1BBL: 4-1BB ligand, TGFb-1: tumour necrosis factor β1, PD-1: Programmed cell death protein 1, PD-L1: Programmed Death-ligand 1.

**Table 1 cells-10-00538-t001:** Main actions of IL-2, IL-7, IL15 & IL-21 on memory-like CD8^+^ T cells.

	IL-2	IL-7	IL-15	IL-21
Survival		X	X	
Memory differentiation:				
Central memory	X ^†^	X		
Effector memory	X ^§^		X	X
Homeostatic proliferation		X	X	
Ag-specific proliferation	X		X	X
Decrease of negative IC	X	X		X
Long-lasting stem-like cells			X	X
Self-renewal			X	
Catabolic mitochondrial reprogramming			X	X

^§^ Low dose, autocrine, ^†^ High dose, paracrine.

## Data Availability

Not applicable.

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
