# Peer review of "Gamma-Chain Receptor Cytokines & PD-1 Manipulation to Restore HCV-Specific CD8+ T Cell Response during Chronic Hepatitis C"

_cells, 2021, doi:10.3390/cells10030538_

Round 1

Reviewer 1 Report

Peña-Asensio et al. describe here the molecular pathways of CD8 T cell exhaustion during chronic HCV infection, and whether combination of both PD-1 blockage and gamma-chain cytokines could contribute to restore T cell functionality and improve infection control. Although the combination of immune checkpoint inhibitors and gamma-chain (γc) cytokines is unlikely to be applied in HCV-infected patients, it represents a promising therapeutic approach for other chronic infectious diseases.

This review gets insight into the exhaustion program of CD8 T cells during chronic antigen stimulation, especially focusing in the role of PD-1 as a potent inhibitor of TCR-dependent activation of certain T cell subsets. Furthermore, the argument for the beneficial synergistic effect of anti-PD-1 and γc cytokine therapies to restore T cell reactivity is solid and sustained by basic immunology concepts and in vitro and in vivo data. The manuscript is clearly written and quite straightforward, although some ideas and references are a little bit repetitive. Therefore, I would recommend that the authors go the extra mile to avoid redundancies and to make the review flow better.

Major comments:

1. The introduction and first section provide useful information for the readers. Nevertheless, some information presented is not completely accurate or well-described.

  • In the introduction (lines 56-58, page 2), authors should also mention that extrinsic regulatory pathways are also involved in HCV-specific CD8+ T cell exhaustion (i.e., lack of CD4+ T cell help and functional suppression by regulatory CD4+ T cells).
  • In section 2 (lines 79-80, page 2), authors assume that HCV-specific CD8+ T cells appear soon after HCV detection. However, HCV-specific CD8+ T cells become detectable in the circulation a few weeks after infection (e.g. ref 5, 31). I would recommend clarifying this statement.
  • In section 2 (lines 99-101, pages 2 and 3), the authors should denote that there are multiple mechanisms involved in HCV persistence apart from escape mutations and T cell exhaustion (e.g., inhibition of IFN-signalling molecules by HCV proteins).
  • In lines 105-107 (page 3), authors briefly mention the hierarchical pattern of T cell exhaustion. To further make clear this process, I would suggest describing it shortly and even including a figure. Since the review is all about potential strategies to restore T cell response, the inclusion of a didactic definition of exhaustion will certainly help readers who are not fully familiar with the topic.
  • At the end of section 2 (lines 151-153, page 4), I would recommend the authors to replace the term “positive regulator” with a rigorous definition to avoid any confusion (e.g., these cells promote the effector functions of exhausted CD8 T cells).

2. In section 3, the authors provide a general view of the inhibitory potency of PD-1 against T cell activation upon TCR stimulation, and some points should be discussed:

  • It has been recently reported that PD-1 can inhibit functional T cell activation upon TCR stimulation in the absence as well as in the presence of CD28 co-stimulation (Mitzuno R et al. Front Immunol. 2019 Mar 29;10:630. doi: 10.3389/fimmu.2019.00630. ). Moreover, the presence of Tregs expressing CTLA-4 can degrade CD80 and CD86 ligands influencing CD28 co-stimulation (Qureshi OS et al. Science. 2011 Apr 29;332(6029):600-3. doi: 10.1126/science.1202947), and therefore, potentially, PD-1 blocking efficacy. It would be interesting to discuss these observations.
  • Considering the complexity of TCR-mediated signalling pathways for T cell activation, the authors should discuss whether alternative mechanisms could sustain TCR signalling and therefore T cell activation, in the absence of CD28 co-stimulation.
  • In Figure 2, the authors should include B7-H2 together with CD80/CD86 molecules. Binding of B7-H2 to CD28 appears to induce a very similar, if not identical, co-stimulatory signal upon TCR co-ligation as the interaction of CD80/CD86 and CD28 (Yao S et al. Immunity. 2011 May 27;34(5):729-40. doi: 10.1016/j.immuni.2011.03.014).

3. In section 4, the authors describe the role of possible combinations of IL-7, IL-15 and IL-21 plus PD-1 blockade to revert T cell exhaustion. In this sense, two key aspects should be discussed:

  • IL-2 is the only FDA-approved γc cytokine to treat patients with cancer. It would be worthwhile to discuss combination therapies of IL-2 anti-PD-1/anti-PD-L1 antibodies.
  • The possible effect of cytokines to mediate a systemic pro-inflammatory state and the potential beneficial/adverse impact in the host.

4. Although the authors greatly summarized the only in vitro study assessing anti-PD-1 and γc cytokine treatment on HCV-specific CD8+ T cells, it would reinforce the possible efficacy of this treatment to mention similar studies in other viral infections.

5. However, results from different studies highlighted the ambiguous role of PD-1 in defining an ineffective T cell response (e.g. Cell. (2016) 8;166(6):1500-1511), and the authors should make a clear statement about that.

Minor comments:

  • In page 2 (lines 63-69), it should be clarified that the paragraph refers to CD8 T cells. This should be checked throughout the manuscript, as some of the references include data for CD4 and CD8 T cells, and only results for CD8 T cells are mostly highlighted.
  • Authors should correct “PD-“ (line 112, page 3).
  • I would recommend the authors to change “by the expression of more” for “by upregulated expression of” (line 116, page 3).
  • In lines 168 and 169 (page 4), I would suggest the authors to replace “PD-1 negatively signals by preferentially dephosphorylating CD28” with “PD-1 suppresses T cell function by preferentially dephosphorylating CD28”.
  • Page 5, line 219: gamma (γ) is missing from the title “4. Gamma chain cytokines for T cell exhaustion reversion”.
  • Page 6, lines 262-263: The following statement “but this cytokine is also key in dictating the composition of the specific T cell pool [72]”, although referenced, is quite vague. I would suggest elaborating it by explaining how IL-15 dictates the composition of the virus-specific T cell pool. Otherwise, better skip it.

Author Response

Reviewer #1

Peña-Asensio et al. describe here the molecular pathways of CD8 T cell exhaustion during chronic HCV infection, and whether combination of both PD-1 blockage and gamma-chain cytokines could contribute to restore T cell functionality and improve infection control. Although the combination of immune checkpoint inhibitors and gamma-chain (γc) cytokines is unlikely to be applied in HCV-infected patients, it represents a promising therapeutic approach for other chronic infectious diseases.

This review gets insight into the exhaustion program of CD8 T cells during chronic antigen stimulation, especially focusing in the role of PD-1 as a potent inhibitor of TCR-dependent activation of certain T cell subsets. Furthermore, the argument for the beneficial synergistic effect of anti-PD-1 and γc cytokine therapies to restore T cell reactivity is solid and sustained by basic immunology concepts and in vitro and in vivo data. The manuscript is clearly written and quite straightforward, although some ideas and references are a little bit repetitive. Therefore, I would recommend that the authors go the extra mile to avoid redundancies and to make the review flow better.

Major comments:

  1. The introduction and first section provide useful information for the readers. Nevertheless, some information presented is not completely accurate or well-described.

In the introduction (lines 56-58, page 2), authors should also mention that extrinsic regulatory pathways are also involved in HCV-specific CD8+ T cell exhaustion (i.e., lack of CD4+ T cell help and functional suppression by regulatory CD4+ T cells).

Response: According to reviewer’s suggestion we have also listed the loss of CD4 T help and the induction of Tregs as mechanisms involved in the development of CD8+ T cell exhaustion (lines 60-61, page 2).

In section 2 (lines 79-80, page 2), authors assume that HCV-specific CD8+ T cells appear soon after HCV detection. However, HCV-specific CD8+ T cells become detectable in the circulation a few weeks after infection (e.g. ref 5, 31). I would recommend clarifying this statement.

Response: We have clarified this statement, highlighting that there is a strong correlation between the appearance of interferon-g secreting HCV-specific CD8+ T cells and viral clearance (lines 87-89, page 2)

In section 2 (lines 99-101, pages 2 and 3), the authors should denote that there are multiple mechanisms involved in HCV persistence apart from escape mutations and T cell exhaustion (e.g., inhibition of IFN-signalling molecules by HCV proteins).

Response: although all these HCV escape mechanisms are out of the scope of this review, we agree with the reviewer that these should be listed. We have added some lines (80-86, page 2) to mention all the previously described HCV mechanisms to escape from immune response and we have cited some reviews about these topics.

In lines 105-107 (page 3), authors briefly mention the hierarchical pattern of T cell exhaustion. To further make clear this process, I would suggest describing it shortly and even including a figure. Since the review is all about potential strategies to restore T cell response, the inclusion of a didactic definition of exhaustion will certainly help readers who are not fully familiar with the topic.

Response: following the reviewer’s suggestion we have added an initial definition of exhaustion (lines 113-114) and we have described in more detail the hierarchical exhaustion process (lines 114-20). We have also added an illustrative figure of this process (Figure 1).

At the end of section 2 (lines 151-153, page 4), I would recommend the authors to replace the term “positive regulator” with a rigorous definition to avoid any confusion (e.g., these cells promote the effector functions of exhausted CD8 T cells).

Response: we have rephrased the sentence to show that the presence of TCF1+ specific CD8+ T cells promotes the effector functions of exhausted CD8 T cells, as indicated by the reviewer (lines 170-171, page 4).

  1. In section 3, the authors provide a general view of the inhibitory potency of PD-1 against T cell activation upon TCR stimulation, and some points should be discussed:

It has been recently reported that PD-1 can inhibit functional T cell activation upon TCR stimulation in the absence as well as in the presence of CD28 co-stimulation (Mitzuno R et al. Front Immunol. 2019 Mar 29;10:630. doi: 10.3389/fimmu.2019.00630.). Moreover, the presence of Tregs expressing CTLA-4 can degrade CD80 and CD86 ligands influencing CD28 co-stimulation (Qureshi OS et al. Science. 2011 Apr 29;332(6029):600-3. doi: 10.1126/science.1202947), and therefore, potentially, PD-1 blocking efficacy. It would be interesting to discuss these observations.

Response: According to reviewer’s suggestion, we have explained that PD-1 could also inhibit TCR triggering in a non-CD28 dependent manner. We also speculate that the blockade of this PD-1 mechanism could be useful as a short-term strategy to rescue effector cells, but to get an extended response it will probable necessary to restore CD28 positive memory cells (Lines 197-202, page 4 and 412-415, page 9).

We have also discussed the potential influence of CTLA-4 expressing T regs on the response of CD8+ T cells to PD-1 blockade, since CTLA-4 could degrade the CD28 ligands, which could impair the T cell restoration after anti-PD-1 treatment (Lines 202-208, page 5).

Considering the complexity of TCR-mediated signalling pathways for T cell activation, the authors should discuss whether alternative mechanisms could sustain TCR signalling and therefore T cell activation, in the absence of CD28 co-stimulation.

Response: we have stated that in those CD28 negative effector CD8 T cells other late positive co-stimulatory molecules such as 4-1BB o GITR are up-regulated to sustain T cell activation, although these co-stimulatory pathways can become dysfunctional during persistent HCV infection (Lines 208-14, page 5).

In Figure 2, the authors should include B7-H2 together with CD80/CD86 molecules. Binding of B7-H2 to CD28 appears to induce a very similar, if not identical, co-stimulatory signal upon TCR co-ligation as the interaction of CD80/CD86 and CD28 (Yao S et al. Immunity. 2011 May 27;34(5):729-40. doi: 10.1016/j.immuni.2011.03.014).

Response: we have added “B7-H2” to the legend of the symbol representing the CD28 ligand. Now this figure is the number 3, because we have added a new figure 1 to explain the exhaustion process.

  1. In section 4, the authors describe the role of possible combinations of IL-7, IL-15 and IL-21 plus PD-1 blockade to revert T cell exhaustion. In this sense, two key aspects should be discussed:

IL-2 is the only FDA-approved γc cytokine to treat patients with cancer. It would be worthwhile to discuss combination therapies of IL-2 anti-PD-1/anti-PD-L1 antibodies.

Response: we have added in section 4 an introductory paragraph about the functions of IL-2 on CD8+ T cells (lines 273-91, page 6). We have also updated the table one with a column highlighting the functions of IL-2 (page 8). Finally, we have discussed two references showing the potential effects of IL-2 plus PD-1/PD-L1 blockade combination, according to IL-2 level (lines 399-408, page 9).

The possible effect of cytokines to mediate a systemic pro-inflammatory state and the potential beneficial/adverse impact in the host.

Response: we have discussed the potential autoimmune and pro-inflammatory adverse events due to potential therapies combining gc cytokines and PD-1/PD-L1 blockade (lines 456-68, page 10).

  1. Although the authors greatly summarized the only in vitro study assessing anti-PD-1 and γc cytokine treatment on HCV-specific CD8+ T cells, it would reinforce the possible efficacy of this treatment to mention similar studies in other viral infections.

Response: we have discussed a similar work in HIV infection that also shows the loss of 4-1BB signal transducer (TRAF1) and the re-expression of this factor by IL-7 treatment and the restoration of T cell function by anti-PD-1 treatment (Exp Med 2012, 209, 77-91, doi:10.1084/jem.20110675), (lines 444-50, page 9).

  1. However, results from different studies highlighted the ambiguous role of PD-1 in defining an ineffective T cell response (e.g. Cell. (2016) 8;166(6):1500-1511), and the authors should make a clear statement about that.

Response: In the section 3 we have added some lines to explain that a high PD-1 level could also mean T cell activation, mainly during the acute phase of HCV infection (lines 183-89, page 4).

Minor comments:

In page 2 (lines 63-69), it should be clarified that the paragraph refers to CD8 T cells. This should be checked throughout the manuscript, as some of the references include data for CD4 and CD8 T cells, and only results for CD8 T cells are mostly highlighted.

Response: we have added CD8+ before T cells in all those cases in which it was omitted, because the authors thought it was not necessary, thinking that it was known by the context.

Authors should correct “PD-“ (line 112, page 3).

Response: this typing error has been corrected.

I would recommend the authors to change “by the expression of more” for “by upregulated expression of” (line 116, page 3).

Response: we have performed this change.

In lines 168 and 169 (page 4), I would suggest the authors to replace “PD-1 negatively signals by preferentially dephosphorylating CD28” with “PD-1 suppresses T cell function by preferentially dephosphorylating CD28”.

Response: we have performed this change

Page 5, line 219: gamma (γ) is missing from the title “4. Gamma chain cytokines for T cell exhaustion reversion”.

Response: we have performed this change

Page 6, lines 262-263: The following statement “but this cytokine is also key in dictating the composition of the specific T cell pool [72]”, although referenced, is quite vague. I would suggest elaborating it by explaining how IL-15 dictates the composition of the virus-specific T cell pool. Otherwise, better skip it.

Response: we have added some information to explain how IL-15 not only sustains memory response, but also regulates its composition (lines 320-25, page 7).

Finally, following reviewer’s advice we have tried to avoid redundancies to make the review flow better

Reviewer 2 Report

In this manuscript, Peña-Asensio and colleagues review immunotherapeutic strategies to improve adaptive immune host response to HCV infection. In particular, how to boost HCV-specific cytotoxic T cell response to avoid virus scape and promote host T cell control. The authors propose the blockade of PD-1/PD-L1 28 checkpoint and addition of gamma-chain receptor cytokines involved in memory generation and maintenance (e.g. IL-7, IL-15, and IL-21) to restored T cell response to HCV infection.

This review is well written and contains interesting data regarding the T cell response to HCV. However, it is unclear for this review the utility of the proposed strategy for a better treatment of HCV infection. As stated by the authors, HCV eradication with current antivirals is being reached in near 100% of the cases, and for the first time we are envisaging the global eradication of HCV. I would suggest the authors to discuss the proposed strategies with other persisting viruses such as HIV and HBV, which are being much more difficult to eradicate. In other words, how we can apply our immune knowledge of HCV infection to other persistent virus infections or other pathologies (e.g. cancer) in which immunotherapies are being widely explored.

Author Response

Reviewer #2

This review is well written and contains interesting data regarding the T cell response to HCV. However, it is unclear for this review the utility of the proposed strategy for a better treatment of HCV infection. As stated by the authors, HCV eradication with current antivirals is being reached in near 100% of the cases, and for the first time we are envisaging the global eradication of HCV. I would suggest the authors to discuss the proposed strategies with other persisting viruses such as HIV and HBV, which are being much more difficult to eradicate. In other words, how we can apply our immune knowledge of HCV infection to other persistent virus infections or other pathologies (e.g. cancer) in which immunotherapies are being widely explored.

Response: we agree with the reviewer that the anti-PD-1 plus gc cytokine strategies plotted in this review are unlikely to be used to treat HCV infection. Nevertheless, this topic was suggested to us by the Guest Editors of the Special Issue: “T and NK Cell-Based Immunotherapy in Chronic Viral Hepatitis and Hepatocellular Carcinoma”, based in our previous expertise.

We also agree that should be interesting to review the role of this immunotherapeutic approach in other human chronic viral infections and cancers but, we think this would be out of the scope of the proposed topic by the Guest Editors. Anyhow, we have also discussed a previous work in restoring HIV-specific CD8 T cell response by combining IL-7 plus anti PD-1 treatment.

Round 2

Reviewer 1 Report

The authors have fully addressed all the issues raised by this reviewer. I would like to acknowledge that the authors have done a great job gathering all the relevant information and presenting it in a structured and very didactic way.